# NATURAL LANGUAGE INFERENCE IMPROVES COMPOSITIONALITY IN VISION-LANGUAGE MODELS

**Paola Cascante-Bonilla** [1,2]  **Yu Hou**[1]  **Yang Trista Cao**[3]  **Hal Daumé III**[1]  **Rachel Rudinger**[1]
[1]University of Maryland, College Park  [2]Stony Brook University  [3]University of Texas at Austin

## ABSTRACT

Compositional reasoning in Vision-Language Models (VLMs) remains challenging as these models often struggle to relate objects, attributes, and spatial relationships. Recent methods aim to address these limitations by relying on the semantics of the textual description, using Large Language Models (LLMs) to break them down into subsets of questions and answers. However, these methods primarily operate on the surface level, failing to incorporate deeper lexical understanding while introducing incorrect assumptions generated by the LLM. In response to these issues, we present Caption Expansion with Contradictions and Entailments (CECE), a principled approach that leverages Natural Language Inference (NLI) to generate entailments and contradictions from a given premise. CECE produces lexically diverse sentences while maintaining their core meaning. Through extensive experiments, we show that CECE enhances interpretability and reduces overreliance on biased or superficial features. By balancing CECE along the original premise, we achieve significant improvements over previous methods without requiring additional fine-tuning, producing state-of-the-art results on benchmarks that score agreement with human judgments for image-text alignment, and achieving an increase in performance on Winoground of $+19.2\%$ (group score) and $+12.9\%$ on EqBen (group score) over the best prior work (finetuned with targeted data). Project page: https://cece-vlm.github.io/

## 1 INTRODUCTION

Trained with internet-scale data, Large-scale Vision-Language Models (VLMs) often struggle to relate objects, attributes, understand spatial relationships, and grasp subtle changes in meaning due to small variations in images or word order (Thrush et al., 2022; Diwan et al., 2022; Wang et al., 2023b; Bitton-Guetta et al., 2023; Yuksekgonul et al., 2023; Tong et al., 2024; Saxon et al., 2024; Fu et al., 2024). With a seeming inability to handle semantically modular scenarios, their opaque nature makes it difficult to understand their decision-making processes (Dziri et al., 2023; Kamath et al., 2023; 2024). Furthermore, internal biases play a major role in affecting the model's performance across various tasks (Zhou et al., 2022; Tiong et al., 2024; Howard et al., 2024; Fraser & Kiritchenko, 2024; Raj et al., 2024). Recent works have explored ways to mitigate these issues by breaking down a problem into smaller tasks. Typically, a Large Language Model (LLM) is prompted to create small programs (i.e., Visual Programming (VP) (Gupta & Kembhavi, 2023; Hu et al., 2023; Surís et al., 2023; Subramanian et al., 2023; Cho et al., 2023b; Koo et al., 2024; Hu et al., 2024b)) or deconstruct the textual description in a set of validation questions with their corresponding expected answers (i.e., Sentence Decomposition via Semantics (SDS) (Cho et al., 2023a; Wu et al., 2023; Yarom et al., 2023; Mitra et al., 2024; Zhang et al., 2024; Wan et al., 2024)). While these methods provide interpretability, they also tend to degrade the VLM performance when evaluating challenging benchmarks that introduce pairs of images and captions that require extensive real-world knowledge and reasoning (Lin et al., 2024).

Consider Figure 1; the first block shows two images with bananas. If we use a VLM to compute the likelihood of answering *"yes"* given each image and the text *"there is a banana split"*, the VLM incorrectly assigns a higher probability to the second image. When decomposing the sentence through SDS, the LLM will output sentences like: *"there is a banana"* and *"the banana is split"* (output examples taken from Cho et al. (2023a)). Although the SDS decomposition seems correct, it preserves the same lexical surface of the text and is unable to incorporate additional information

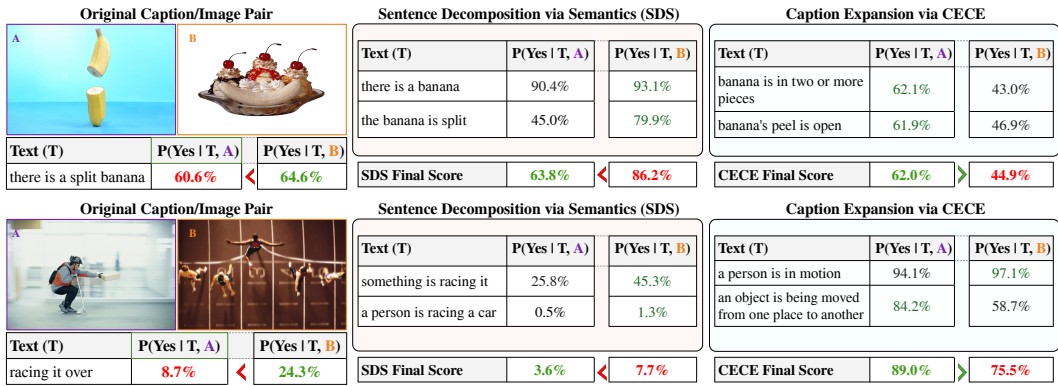

Figure 1: Examples from Winoground dataset. The first column shows the output of LLaVa-1.6 when computing the likelihood of answering *"yes"* given the image and text. The second column shows the sentence decomposition proposed in prior work (SDS), which follows the original caption semantics. The third column shows our proposed Caption Expansion with Contradictions and Entailments (CECE). In all cases, the model is only allowed to evaluate one image and text at a time.

that could be leveraged by the VLM to make a correct prediction – both images contain bananas, but the VLM might have learned a stronger correlation of a *banana split* being a *dessert*. Likewise, if we look at the second block in Figure 1 and follow the same process, the SDS output will not only preserve the same lexical surface, but will make wrong assumptions and introduce some of the biases present in the LLM (e.g., *racing* might have strong correlations with *cars*).

To address these limitations, we introduce **Caption Expansion with Contradictions and Entailments (CECE)**, a principled approach that leverages Natural Language Inference (NLI) to improve the compositional capabilities of VLMs. With CECE, we instruct an LLM to use NLI – which is used to determine the relationship between two sentences, a premise and a hypothesis (Bowman et al., 2015) –, and generate entailments (hypotheses that logically follow from the premise) and contradictions (hypotheses that are logically incompatible with the premise). In this way, the LLM is instructed to produce lexically diverse sentences while preserving the underlying meaning of the original captions. It is important to note that while the outputs generated by SDS can be considered a subset of entailments, CECE expands the scope by generating a wider variety of captions that capture nuanced relationships beyond what SDS would typically output, including both entailments and contradictions. This allows the LLM to leverage common sense and break down the captions using world knowledge, while also mitigating the hallucinations and biases introduced by the LLM.

Revisiting Figure 1, CECE generates sentences that are lexically different from the original caption but preserve its meaning (e.g., *"split"* entails dividing things *"in one or more pieces"*). By expanding the captions with both entailments and contradictions, the LLM incorporates world knowledge and common sense while mitigating hallucinations and biases. For example, *"racing it over"* is parsed into *"a person in motion"* and *"an object being moved from one place to another"*, which entails a different perceptual inference of the caption, and reduces the likelihood of stereotypical associations. As a whole, CECE goes beyond the lexical boundaries of SDS, introducing richer contextual information for improved reasoning.

To leverage both entailments and contradictions generated via CECE, we evaluate the likelihood of the VLM answering *"yes"* given the image and each entailment, and the likelihood of the VLM answering *"no"* given the image and each contradiction. These scores are then aggregated using a weighting value to balance the contributions of both entailments and contradictions. With CECE, we incorporate both positive and negative reasoning cues: entailments provide semantic inclusion (focusing on subset relations), while contradictions provide semantic exclusion (focusing on subset complements). In addition, we aggregate the VLM likelihood of answering *"yes"* given the image and original caption. We found that including the original caption provides additional context for balancing out the VLM outputs. In a way, the original caption serves as a direct reference that helps the model minimize the risk of semantic drift, where caption expansions may diverge from the intended meaning. Our results show that incorporating CECE along with the original caption further improves the image-to-text and text-to-image alignment, providing lexical diversity, improved semantic reasoning, and a more interpretable output, without fine-tuning the models, which may compromise their zero-shot capabilities.

Our contributions are summarized as follows: a) We propose Caption Expansion with Contradictions and Entailments (CECE), a principled approach that leverages entailments and contradictions to preserve the semantic meaning while providing lexical diversity of text descriptions. b) We show that CECE significantly outperforms prior decomposition methods, obtaining $47.5\%$ on Winoground (group score) and $47.9\%$ on EqBen (group score) without finetuning. c) We conduct extensive experiments on benchmarks that score agreement with human judgments of alignment for image-text alignment. d) We provide thorough ablation studies and analyses to evaluate the performance of our method under various conditions, and introduce a simple ensembling approach that effectively boosts the accuracy when associating each image-text pair.

## 2 RELATED WORK

**Single-caption Scoring frameworks.** Commonly used for evaluating the alignment between text and images in VLMs (Dai et al., 2023; Liu et al., 2023b;a; 2024; Bai et al., 2023), these approaches include similarity scores derived from multimodal encoders (Radford et al., 2021), as well as text similarity metrics based on image captioning models (Li et al., 2023). However, summarizing the relationship between text and images using single embeddings often fails to capture the semantic granularity needed for fine-grained image-text alignment (Zhao et al., 2024). Moreover, these metrics are often uncalibrated and may obscure important nuances; for example, a particular CLIPScore value might indicate a good match for pixel art but be considered poor for realistic images, which might affect image-text alignment compared to the human judgments of alignment in text-to-image evaluation metrics. More recently, Lin et al. (2024) introduced VQAScore, which leverages a VLM to computes the likelihood of a given image-caption pair, by re-writing the caption as a binary question ("yes|no"), yielding significant improvements. Our approach builds on this while aiming to provide a more detailed and semantically diverse evaluation. By generating entailments and contradictions, we introduce a mechanism to understand the model's nuanced response to positive and negative cues, thereby addressing some limitations of prior single-embedding scoring methods.

**Sentence Decomposition via Semantics (SDS) frameworks.** Given the limitations of single-caption scoring frameworks, recent works have explored more sophisticated evaluation methods based on sentence decomposition and semantic analysis (Khan et al., 2023; Hu et al., 2023; Cho et al., 2023b). These approaches aim to provide a more comprehensive and fine-grained evaluation of text-to-image and image-to-text alignment. Typically, an LLM is instructed to generate subsets of validation questions and expected answers that a VLM can evaluate (Wan et al., 2024). Similarly, Sanders et al. (2024) uses entailments to improve video question answering. While these methods provide finer semantic analysis and interpretability to the evaluation process, SDS methods typically produce outputs that are direct entailments of the original caption. Furthermore, Yarom et al. (2023) introduces VNLI, an approach that finetunes a model that receives an image and a set of entailments and contradictions, where the contradictions are defined as identified question-answer pairs with the lowest VQA score. We instead directly instruct the LLM to output entailments and contradictions with the input caption as a premise, providing a strong prior for caption expansion and exclusion, without finetuning the models.

**Chain-of-Thought Prompting frameworks.** Recent work has shown promising results when incorporating Chain-of-Thought (CoT) prompting (Wei et al., 2022) to enhance compositional reasoning in challenging vision-language scenarios, using large-VLMs (Dubey et al., 2024; Achiam et al., 2023; Hu et al., 2024a). Notably, Zhang et al. (2024) introduces CoT for multiple image-to-text matching, through a contrastive approach for comparative reasoning. Similarly, a two-step prompting strategy is introduced to generate descriptions of the given image, which is then used by the model to answer specific questions (Wu et al., 2023; Ossowski et al., 2024). These works prompt the VLM with multiple images and instruct them to choose the correct one. Mitra et al. (2024) propose to generate scene graphs as an intermediate reasoning step, and instruct the VLM to pick from two given captions given the image and scene graph. This also reformulates the problem as a multiple-image-to-text matching task. In contrast, our approach is only allowed to evaluate one image and text at a time – we argue that this setup closely aligns with real-world scenarios, where compositional understanding must be robust without the benefit of direct comparisons between multiple images or captions. By evaluating each image-caption pair independently, we ensure that the model's reasoning is not influenced by relative comparisons, thus providing a more realistic assessment of its compositional capabilities.

## 3 METHODOLOGY

To effectively address the limitations of Vision-Language Models (VLMs) in handling complex compositional visual-textual relationships, we introduce CECE: Caption Expansion with Contradictions and Entailments. Our approach leverages Natural Language Inference (NLI) to systematically generate entailments and contradictions for each image-caption pair, capturing the deeper meaning of the text and a more interpretable metric for image-to-text and text-to-image evaluation and alignment. We describe our proposed method, beginning with the generation of entailments and contradictions via CECE (Section 3.1). We then explain the likelihood computation for each caption expansion (Section 3.2). Finally, we describe the score-balancing mechanism we use to integrate the contributions of entailments, contradictions, and the original captions into a unified evaluation framework (Section 3.3).

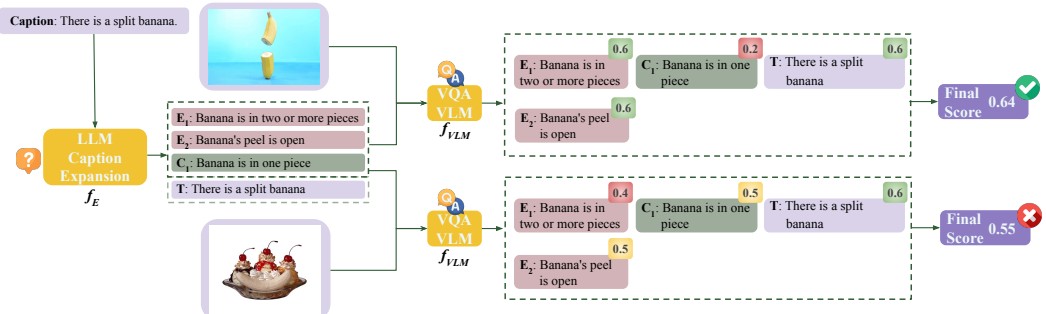

Figure 2: Complete pipeline of our proposed approach. CECE provides diverse semantic inclusion and exclusion given a caption premise. A VLM is then used to compute the likelihood of the captions generated via entailments and contradictions. The scores are finally balanced along with the original image-text results to avoid semantic drift and enable better alignment.

Given an evaluation dataset that consists of image-caption pairs, we define: $X = (T, I)$ as an image-caption pair, where $T$ represents the textual caption and $I$ is the image to evaluate.

### 3.1 CAPTION EXPANSION

To enrich the semantic representation via caption expansion, we instruct an LLM, denoted as $f_E$, to generate entailments and contradictions (i.e. expansions) based on the original caption $T$ (see Figure 2). This process outputs two subsets: the **entailment set** $E$, consisting of hypotheses that logically follow from $T$, and the **contradiction set** $C$, consisting of hypotheses that are logically incompatible with $T$ as follows: $(E, C) = f_E(T)$. Here, $E$ represents the subset of generated entailments: $E = \{e_1, e_2, \ldots, e_n\}$, and $C$ represents the subset of generated contradictions: $C = \{c_1, c_2, \ldots, c_n\}$. Each element in $E$ and $C$ aims to provide a diverse but grounded derivation of the original caption through semantic inclusion and exclusion. [1]

### 3.2 LIKELIHOOD COMPUTATION

Following Lin et al. (2024), we use a Vision-Language Model (VLM), denoted as $f_{\text{VLM}}$, to evaluate image-text pairs. To standardize the input for the VLM, we use a function $q(.)$ that converts a given text $t$ into a question format, which enables the model to assess the visual-textual alignment in a consistent manner. Specifically, $q(.)$ generates a yes/no question that captures the essence of the original statement. If we take the example in Figure 2, the caption $t = $ "THERE IS A BANANA SPLIT" is formatted as $q(t) = $ *"Does* THERE IS A BANANA SPLIT *can be observed in the image? Answer yes or no"*.

We proceed to evaluate all captions generated in the previous step as follows:

---

[1]Prompt details are in Appendix A.1.

(1) For entailments, we compute the likelihood of answering "yes" given the image-text pair:

$$f_{\text{VLM}}^{ent}(e_i, I) = P(\text{"yes"} \mid I, q(e_i)) \tag{1}$$

(2) For contradictions, we compute the likelihood of answering "no" given the image-text pair:

$$f_{\text{VLM}}^{cnt}(c_i, I) = P(\text{"no"} \mid I, q(c_i)) \tag{2}$$

(3) For the original caption, we compute the likelihood of answering "yes" as follows:

$$f_{\text{VLM}}^{cap}(T, I) = P(\text{"yes"} \mid I, q(T)) \tag{3}$$

This step allows us to assess the model's agreement with both positive cues (entailments) and negative cues (contradictions), as well as its consistency with the original caption. In all cases, the probabilities are normalized such that $P(\text{"yes"}) + P(\text{"no"}) = 1$. By evaluating these three components, we better assess the VLM's ability to align visual content with textual information, capturing nuanced relationships across different semantic variations.

## 3.3 Balancing Scores

To integrate the information from entailments, contradictions, and the original caption, we employ a two-step balancing process using hyperparameters $\alpha_1$ and $\alpha_2$, to obtain a more comprehensive evaluation of the model's performance while avoiding semantic drift. By carefully balancing the contributions from entailments, contradictions, and the original caption, this approach ensures that the final assessment remains grounded in the intended meaning of the original text, minimizing the risk of unintended shifts in interpretation that could arise from generated variations (refer to the nuanced scores to the right in Figure 2).

First, we define the aggregation function $S(.)$, which computes the average score across all elements in a given set. For example, to score all entailment outputs given the set $E$ containing $M$ elements, the aggregation function is given by:

$$S(E, I) = \frac{1}{M} \sum_{i=1}^{M} f_{\text{VLM}}^{ent}(e_i, I) \tag{4}$$

and score all contradiction outputs in a similar way:

$$S(C, I) = \frac{1}{M} \sum_{i=1}^{M} f_{\text{VLM}}^{cnt}(c_i, I) \tag{5}$$

Then, we compute a weighted sum of the entailment, contradiction, and the original caption score:

$$S(X) = \alpha_2 \cdot [\alpha_1 \cdot S(E, I) + (1 - \alpha_1) \cdot S(C, I)] + (1 - \alpha_2) \cdot S(T, I) \tag{6}$$

where $S(T, I)$ represents the VLM score for the original image-caption pair.

This two-step mechanism allows for a flexible adjustment of the importance assigned to entailments, contradictions, and the original caption in the final assessment, balancing their contributions. By effectively incorporating both positive and negative reasoning cues along with the original context, we aim to achieve a nuanced evaluation that better aligns with human judgments, avoids semantic drifts, and reduces overreliance on biased or superficial features.

## 4 Experiment Settings

**Baselines.** We compare CECE with a wide range of baseline methods divided into three categories. 1) End-to-end models (i.e., Single-caption Scoring frameworks) including CLIPScore (Radford et al., 2021), BLIP2$_{\text{ITM}}$ (Li et al., 2023), VQAScore (Lin et al., 2024), VIEScore (Ku et al., 2023), and GPT4V-Eval (Zhang et al., 2023); 2) Visual Programming (VP) frameworks (VisProg (Gupta & Kembhavi, 2023), ViperGPT (Surís et al., 2023), VPEval (Cho et al., 2023b)); 3) Sentence Decomposition via Semantic (SDS) frameworks (VQ2 Yarom et al. (2023), DSG (Cho et al., 2023b), CCoT (Mitra et al., 2024)). Note that VP and SDS frameworks use an LLM for program instruction or sentence decomposition, including ChatGPT (OpenAI, 2023), GPT4 (Achiam et al., 2023),

FlanT5 (Chung et al., 2024), and several VLMs, including ViLT (Kim et al., 2021), OWL-ViT (Minderer et al., 2022), CLIP (Radford et al., 2021), GLIP (Li et al., 2022), GroundDINO (Liu et al., 2023c), LLaVA-1.5 (Liu et al., 2023a), LLaVA-1.6 (Liu et al., 2024), PaLI-17B (Chen et al., 2022) and the finetuned model introduced by Lin et al. (2024) CLIP-FlanT5-11B.

**Implementation.** We use Llama3.1 70B (Dubey et al., 2024) as our LLM for caption expansion through NLI. We further evaluate CECE on different VLMs (BLIPv2 (Li et al., 2023), InstructBLIP (Dai et al., 2023), LLaVA-1.5 (Liu et al., 2023a), LLaVA-1.6 (Liu et al., 2024)) and incorporate a soft-assembling method that balances the results scores of different models, by balancing the scores from entailments and contradictions (VLM scores from LLaVA-1.5), and the original caption (VLM scores from LLaVA-1.6). We use $\alpha_1 = 0.5$ and $\alpha_2 = 0.6$ in all experiments. Additional details are included in Section 6.

**Tasks and Benchmarks.** We evaluate the compositional capabilities of CECE in three different tasks. 1) Image-text matching evaluation through binary retrieval tasks, which require determining the best caption from a pair of candidates for a given image, as well as determining the best image from a pair of candidates for a given caption. We report results on two benchmarks (Winoground (Thrush et al., 2022), EqBen (Wang et al., 2023b)) and the performance is evaluated using three metrics: (i) a text score, which assesses the model's ability to identify the correct caption for a given image; (ii) an image score, which measures the model's accuracy in selecting the appropriate image based on a provided caption; and (iii) a group score, which evaluates the successful matching of both pairs. 2) Score agreement with human judgments of alignment for image-text alignment, using images generated from complex text prompts. We report results on five text-to-image evaluation benchmarks (DrawBench (Saharia et al., 2022), EditBench (Wang et al., 2023a), COCO-T2I (Lin et al., 2014), TIFA160 (Hu et al., 2023), Pick-a-Pic (Kirstain et al., 2023)). 3) 3D alignment, which assesses the human judgments of alignment for text-to-3D asset generation. We report results on the StanfordT23D (Wu et al., 2024) benchmark with the human ratings collected by Lin et al. (2024).

Table 1: Performance on challenging compositional benchmarks that require multi-hop reasoning. *Tools* indicate the vision and language models used for inference. *LLM* indicates the large language model used for generating the visual programming output or sentence decompositions. *DSG†* is the only method that uses a model fine-tuned for this task. Llama3.1† indicates the 8B parameter model.

| Method | Tools-$f_{VLM}$ | LLM-$f_E$ | Winoground | | | EqBen | | |
|---|---|---|---|---|---|---|---|---|
| | | | **Text** | **Image** | **Group** | **Text** | **Image** | **Group** |
| Random Chance | – | – | 25.0 | 25.0 | 16.7 | 25.0 | 25.0 | 16.7 |
| Human Evaluation | – | – | 89.5 | 88.5 | 85.5 | – | – | – |
| *End-to-end models* | | | | | | | | |
| CLIPScore (Radford et al., 2021) | CLIP-L-14 | – | 27.8 | 11.5 | 7.8 | 35.0 | 35.0 | 25.0 |
| BLIP2$_{ITM}$ (Li et al., 2023) | BLIPv2 | – | 42.8 | 22.0 | 18.3 | 48.6 | 43.6 | 35.0 |
| VQAScore (Lin et al., 2024) | InstructBLIP | – | 44.5 | 42.8 | 28.5 | 49.3 | 58.6 | 38.6 |
| VQAScore (Lin et al., 2024) | LLaVA-1.5 | – | 45.5 | 41.3 | 29.8 | 45.0 | 47.1 | 28.6 |
| VQAScore (Lin et al., 2024) | LLaVA-1.6 | – | 46.8 | 45.8 | 31.3 | 46.4 | 54.3 | 32.9 |
| VIEScore (Ku et al., 2023) | GPT4-Vision | – | 40.8 | 39.3 | 34.5 | 40.0 | 34.3 | 32.9 |
| GPT4V-Eval (Zhang et al., 2023) | GPT4-Vision | – | 44.5 | 49.0 | 36.3 | 42.9 | 40.0 | 35.0 |
| *Visual Programming (VP)* | | | | | | | | |
| VisProg (Gupta & Kembhavi, 2023) | ViLT, OWL-ViT | ChatGPT | 3.5 | 3.5 | 3.5 | 7.9 | 7.9 | 7.9 |
| ViperGPT (Surís et al., 2023) | CLIP, BLIP, GLIP | ChatGPT | 7.8 | 7.8 | 7.8 | 4.3 | 4.3 | 4.3 |
| VPEval (Cho et al., 2023b) | BLIPv2, GroundDINO | ChatGPT | 12.8 | 11.0 | 6.3 | 34.3 | 25.7 | 21.4 |
| *Sentence Decomposition via Semantics (SDS)* | | | | | | | | |
| DSG (Cho et al., 2023a) | LLaVA-1.5 | Llama3.1† | 5.7 | 9.5 | 3.7 | 10.0 | 14.3 | 6.4 |
| DSG (Cho et al., 2023a) | LLaVA-1.6 | Llama3.1† | 4.5 | 10.2 | 2.7 | 10.7 | 14.3 | 6.4 |
| VQ2 (Yarom et al., 2023) | LLaVA-1.5 | FlanT5 | 14.0 | 27.3 | 10.0 | 22.9 | 40.7 | 20.0 |
| DSG (Cho et al., 2023a) | LLaVA-1.5 | ChatGPT | 21.0 | 16.8 | 15.5 | 26.4 | 20.0 | 20.0 |
| DSG (Cho et al., 2023a) | LLaVA-1.6 | ChatGPT | 45.8 | 45.8 | 31.3 | 47.1 | 44.3 | 32.1 |
| DSG† (Cho et al., 2023a) | CLIP-FlanT5-11B | ChatGPT | 41.0 | 38.3 | 28.3 | 45.7 | 47.9 | 35.0 |
| CCoT (Mitra et al., 2024) | LLaVA-1.5 | GPT4 | 39.8 | 37.3 | 22.3 | – | – | – |
| VQ2 (Yarom et al., 2023) | PaLI-17B | FlanT5 | 47.0 | 42.0 | 30.5 | – | – | – |
| *Caption Expansion with Contradictions and Entailments (*CECE*)* | | | | | | | | |
| CECE (Ours) | BLIPv2 | Llama3.1 | 29.8 | 39.3 | 21.5 | 30.0 | 43.6 | 21.4 |
| CECE (Ours) | InstructBLIP | Llama3.1 | 37.5 | 46.3 | 28.8 | 41.4 | 57.1 | 34.3 |
| CECE (Ours) | LLaVA-1.5 | Llama3.1† | 47.7 | 49.7 | 35.5 | 48.6 | 54.3 | 35.0 |
| CECE (Ours) | LLaVA-1.6 | Llama3.1† | 48.0 | 57.5 | 38.7 | 50.7 | 64.3 | 40.0 |
| CECE (Ours)* | LLaVA-1.5, LLaVA-1.6 | Llama3.1† | 50.0 | 53.5 | 39.0 | 53.6 | 57.1 | 40.7 |
| CECE (Ours) | LLaVA-1.5 | Llama3.1 | 51.3 | 55.3 | 41.0 | **58.6** | 57.9 | 41.4 |
| CECE (Ours) | LLaVA-1.6 | Llama3.1 | 52.0 | 61.3 | 42.8 | **58.6** | 64.3 | 47.1 |
| CECE (Ours)* | LLaVA-1.5, LLaVA-1.6 | Llama3.1 | **55.0** | **61.3** | **47.5** | 57.9 | **65.0** | **47.9** |

## 5 RESULTS

**Image-text matching evaluation through binary retrieval tasks.** We conduct experiments on Winoground and EqBen. Results are shown in Table 1. The entailments and contradictions generated by CECE can be applied to a wide variety of VLMs, this allows for a comprehensive evaluation for a wide range of visual-language model architectures. We demonstrate that our method outperforms prior works (including single-caption scoring methods (i.e, end-to-end models), visual programming, and sentence decomposition approaches that also leverage LLMs). For a fair comparison, we took the best SDS method (DSG) and ran their end-to-end framework using Llama3.1 and LLaVA-1.6. We also run DSG with the finetuned method introduced by Lin et al. (2024). Note that while CECE outperforms all other methods under similar conditions (one-LLM, one-VLM), the best results are obtained through our score-balancing approach CECE*, which leverages both LLaVA-1.5 (scores from entailments and contradictions) and LLaVA-1.6 (scores from the original caption).

Table 2: Performance on benchmarks that score agreement with human judgments of alignment for image-text alignment. *Tools* indicate the vision and language models used for inference. *LLM* indicates the large language model to generate the sentence decompositions. Note that none of the models have been specifically fine-tuned for this task.

| Method | Tools-$f_{VLM}$ | LLM-$f_E$ | DrawBench | EditBench | COCO-T2I | TIFA160 | Pick-a-Pic |
|---|---|---|---|---|---|---|---|
| *End-to-end models* | | | | | | | |
| CLIPScore (Radford et al., 2021) | CLIP-L-14 | – | 49.1 | 60.6 | 63.7 | 54.1 | 76.0 |
| BLIP2$_{ITM}$ (Li et al., 2023) | BLIPv2 | – | 60.5 | 68.0 | 70.7 | 57.5 | 80.0 |
| VQAScore (Lin et al., 2024) | InstructBLIP | – | 82.6 | 75.7 | 83.0 | 70.1 | 83.0 |
| VQAScore (Lin et al., 2024) | LLaVA-1.5 | – | 82.2 | 70.6 | 79.4 | 66.4 | 76.0 |
| VIEScore (Ku et al., 2023) | GPT4-Vision | – | – | – | – | 63.9 | 78.0 |
| GPT4V-Eval (Zhang et al., 2023) | GPT4-Vision | – | – | – | – | 64.0 | 74.0 |
| *Sentence Decomposition via Semantics (SDS)* | | | | | | | |
| VQ2 (Yarom et al., 2023) | LLaVA-1.5 | FlanT5 | 52.8 | 52.8 | 47.7 | 48.7 | 73.0 |
| DSG (Cho et al., 2023a) | LLaVA-1.5 | ChatGPT | 78.8 | 69.0 | 76.2 | 54.3 | 70.0 |
| VQ2 (Yarom et al., 2023) | PaLI-17B | FlanT5 | 82.6 | 73.6 | 83.4 | – | – |
| TIFA (Hu et al., 2023) | PaLI-17B | Llama2 | 73.4 | 67.8 | 72.0 | – | – |
| *Caption Expansion with Contradictions and Entailments (CECE)* | | | | | | | |
| CECE (Ours) | InstructBLIP | Llama3.1 | 85.4 | **76.7** | 81.4 | 69.3 | 84.0 |
| CECE (Ours) | LLaVA-1.5 | Llama3.1 | 87.3 | 75.6 | 81.3 | 68.9 | **86.0** |
| CECE (Ours) | LLaVA-1.6 | Llama3.1 | 86.3 | 75.9 | **83.8** | **70.4** | 83.0 |
| CECE (Ours)* | LLaVA-1.5, LLaVA-1.6 | Llama3.1 | **88.2** | 76.4 | 83.0 | 69.8 | 85.0 |

**Score agreement with human judgments of alignment for image-text alignment.** We show results on five text-to-image evaluation benchmarks in Table 2. These results measure the correlation of each method score with human judgments of alignments for an image generated based on a textual prompt. Human ratings are given on a 1-to-5-Likert scale or by assigning a binary match-or-not label. We show AUROC for DrawBench, EditBench, and COCO-T2I, pairwise accuracy for TIFA160, and binary accuracy for Pick-a-Pick. CECE consistently outperforms all prior scoring approaches, indicating that caption expansion via NLI better aligns with human judgments when evaluating text-to-image generation methods.

Table 3: Performance on benchmarks that correlate 3D alignment with human agreement.

| Method | Pairwise Acc | Pearson | Kendall |
|---|---|---|---|
| *End-to-end models* | | | |
| CLIPScore | 61.0 | 48.1 | 32.6 |
| BLIPv2Score | 56.6 | 34.3 | 23.4 |
| InstructBLIP | 68.0 | 59.5 | 47.5 |
| LLaVA-1.5 | 64.9 | 55.8 | 40.8 |
| *Finetuned on human feedback* | | | |
| ImageReward | 66.3 | 57.1 | 43.6 |
| PickScore | 60.1 | 41.3 | 30.3 |
| HPSv2 | 55.9 | 31.5 | 21.9 |
| *Caption Expansion with Contradictions and Entailments (CECE)* | | | |
| w/ InstructBLIP | **68.5** | **64.0** | **48.4** |
| w/ LLaVA1.5 | 65.3 | 57.4 | 41.8 |

**3D alignment with human agreement.** We show results in Table 3. We report the pairwise accuracy along with the Pearson and Kendall coefficients, which assume a linear correspondence between human ratings and metric scores. We follow the setting proposed by Lin et al. (2024) and show that CECE consistently outperforms the base model (LLaVA-1.5).

# 6 ANALYSIS

We conduct an in-depth analysis of CECE through multiple perspectives, including detailed breakdowns on Winoground and Eqben benchmarks, lexical diversity, semantic drift, and the impact of incorporating entailments and contradictions. Moreover, we demonstrate the robustness of CECE across different model architectures and present a comprehensive ablation study to understand the importance of each component in our approach.

**Detailed results on Winoground.** We show fine-grained results on tags provided by Winoground in Tables 4. Each sample is grouped per skill category and can include multiple skills. We compare our method against the results from the base end-to-end models, since they outperform prior work based on SDS. Notably, CECE not only outperforms objects and relations from the linguistic side, but it also outperforms in cases where the images need to be interpreted non-literally due to idiomatic uses of language in a caption. It also outperforms the base models when a symbolic description must be understood to make a correct prediction (e.g., typically in non-natural images, such as drawings or illustrations).

Table 4: Detailed analysis on Winoground. Results are grouped by linguistic ($_L$) and visual ($_V$) tags. We report results using DSG / CECE with LLaVA-1.6, and DSG* / CECE* with LLaVA-1.5 (for entailments and contradictions) and LLaVA-1.6 (for the given caption).

| Method | Object$_L$ | | | Relation$_L$ | | | Both$_L$ | | | Symbolic$_V$ | | | Pragmatics$_V$ | | |
|---|---|---|---|---|---|---|---|---|---|---|---|---|---|---|---|
| | Text | Image | Group | Text | Image | Group | Text | Image | Group | Text | Image | Group | Text | Image | Group |
| Human | 92.20 | 90.78 | 88.65 | 89.27 | 90.56 | 86.70 | 76.92 | 57.69 | 57.69 | 96.43 | 92.86 | 92.86 | 58.82 | 41.18 | 41.18 |
| InstructBLIP | 42.5 | 49.7 | 27.7 | 34.3 | 33.9 | 20.2 | **65.4** | 38.5 | 34.6 | 31.7 | 21.9 | 14.6 | 25.0 | 29.2 | 8.3 |
| LLaVA-1.5 | 46.1 | 46.8 | 28.4 | 45.1 | 43.8 | 33.9 | 46.1 | 38.5 | 26.9 | 46.3 | 36.6 | 24.4 | 33.3 | 20.8 | 12.5 |
| LLaVA-1.6 | 48.2 | 53.9 | 35.5 | 43.8 | 40.8 | 27.9 | **65.4** | 46.2 | 38.5 | 46.3 | 41.5 | 26.8 | 29.2 | 41.7 | 16.7 |
| LLaVA-1.5+1.6 | 51.7 | 53.9 | 36.2 | 49.3 | 47.2 | 34.3 | 61.5 | 46.1 | 38.5 | **56.1** | 43.9 | 31.7 | 37.5 | 33.3 | 20.8 |
| DSG | 45.4 | 44.0 | 27.6 | 42.9 | 41.2 | 30.9 | 61.5 | 53.8 | 46.1 | 46.3 | 41.5 | 29.3 | 33.3 | 33.3 | 20.8 |
| DSG* | 52.5 | 50.3 | 35.5 | 51.0 | 45.5 | 34.3 | 53.8 | 42.3 | 34.6 | 41.4 | 31.7 | 19.5 | **50.0** | 41.7 | **33.3** |
| CECE (Ours) | 51.8 | 66.0 | 43.3 | 51.5 | **59.2** | 42.5 | 57.7 | 53.8 | 42.3 | 53.7 | **65.9** | 39.0 | 45.8 | **50.0** | 33.3 |
| CECE (Ours)* | **56.7** | **68.8** | **49.7** | **53.6** | 56.7 | **45.9** | 57.7 | **61.5** | **50.0** | 53.7 | 58.5 | **43.9** | 37.5 | 41.7 | 33.3 |

**On the complexity of image-text matching.** A key challenge of Winoground is that the captions are also ambiguous; for example, in the sentence *"it hatched before it was eaten"*, *"it"* could refer either to the egg or to the animal inside the egg. Previously identified by Diwan et al. (2022), we show results on the taxonomy of Winoground schemes in Table 5. We compare LLaVA-1.6 VQAScore outputs and DSG scores against CECE. For a fair comparison, we report the results of DSG and CECE using Llama3.1 and LLaVA-1.6. Notably, DSG outperforms the base model and our method when evaluating samples tagged as `NonCompositional`. These sample pairs are actually not semantically compositional of one another since they do not contain semantic entities. On the other hand, our CECE shows stronger results for all other tags, where a higher score is expected for the correct image-text pairs when the image is difficult to parse (e.g., objects are small or blurry), the wording of the caption makes it difficult to parse (e.g., *"yellow duck shoes on"*), or common-sense reasoning to match the correct image-text pair is required (e.g., *"together hammering something"* vs. *"hammering something together"*). It is important to note that in image-text matching datasets, the captions are syntactically similar, with the key difference of contextual or semantic alterations by swapping objects, relations, or both. Similarly, sentences like *"another organism was harmed by a plant, and that plant broke the organism into pieces"* introduce ambiguity regarding the subject, since *"another organism"* could be interpreted as either an animal or possibly another plant. Results show that CECE is particularly performant also where world knowledge is required.

**Lexical diversity and semantic drift.** We conduct comprehensive experiments to measure the lexical diversity between different semantic decomposition methods (SDS) and CECE. We compute the Jaccard Similarity (Real & Vargas, 1996) between the Winoground caption and the LLM outputs for each technique. Results show a higher similarity between captions generated via DSG (with a score of $0.53$) in comparison with captions generated via CECE (with a score of $0.32$).

Table 5: Breakdown analysis with Winoground categories from Diwan et al. (2022). For fair comparison, we report numbers for DSG and CECE under similar conditions (i.e., same LLM and VLM).

| | LLaVA-1.6 | | | DSG | | | CECE | | |
|---|---|---|---|---|---|---|---|---|---|
| | **Text** | **Image** | **Group** | **Text** | **Image** | **Group** | **Text** | **Image** | **Group** |
| *Non Comp.* | 60.0 | 50.0 | 40.0 | **66.7** | **53.3** | **46.7** | **66.7** | **53.3** | 43.3 |
| *Ambig. Correct* | 30.4 | 28.3 | 17.4 | 34.8 | 34.8 | 21.3 | **41.3** | **43.5** | **26.1** |
| *Visually Difficult* | **47.4** | 34.2 | 23.7 | 26.3 | 28.9 | 15.8 | 39.5 | **50.0** | **28.9** |
| *Unusual Text* | **40.0** | 32.0 | 26.0 | 32.0 | 38.0 | 22.0 | **40.0** | **50.0** | **34.0** |
| *Unusual Image* | **50.0** | 41.1 | **33.9** | 44.6 | 33.9 | 19.6 | **50.0** | 58.9 | **33.9** |
| *Complex Reasoning* | **32.1** | 32.1 | 19.2 | 28.2 | 28.2 | 16.7 | 29.5 | **46.2** | 20.5 |

**Human Validation on Caption Expansion.** We validate the quality of the entailment and contradiction captions generated by CECE as described in subsection 3.1. We randomly selected 90 samples and manually annotated whether the generated captions could be entailed from the original caption. Using a Likert scale ranging from $1$ (definitely not likely) to $5$ (definitely likely), the entailment captions received an average score of $4.7$, while the contradiction captions received an average score of $1.7$. This indicates that the entailment and contradictions are both accurately generated.

**Semantic drift and balancing scores.** While the lexical diversity CECE provides benefits the image-text and text-image matching, we also observed a level of divergence from the semantics of the original caption. We refer to this as "semantic drift", a phenomenon present in LLMs that describes the degradation of text generation quality. Spataru et al. (2024) defines this as a degree of separation in generation quality. We mitigate this issue by incorporating the balancing score approach described in subsection 3.3. We show in Figure 3 how different $\alpha$ values balance out the contribution between entailments and contradictions ($\alpha_1$) and the given caption ($\alpha_2$), with a visible trend of highest performance in the middle for each scoring metric.

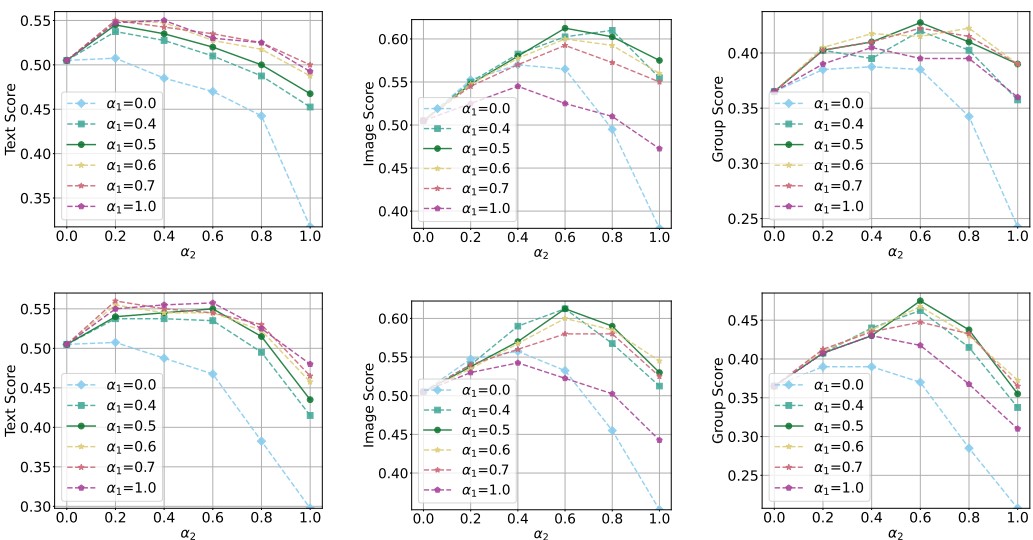

Figure 3: Balancing scores from entailments, contradictions and the original caption. We show how the $\alpha$ values affect the matching performance in Winoground. The first row shows results using LLaVA-1.6 only. The second row shows results using our soft-ensemble approach that balances the scores from LLaVA-1.5 (for entailments and contradictions) and LLaVA-1.6 (for the original caption). Best viewed in color.

**Importance of each component and soft-ensembling.** We ablate each component of our proposed method in three ways. We show in Table 6 results on Winoground and Eqben when using (i) only entailments, (ii) entailments and contradictions, (ii) entailments, contradictions and the given caption. Our results show that entailments alone outperform prior methods, while all components progres-

Table 6: Importance of each component. While entailments alone significantly improve the compositional scoring performance, progressively adding each proposed component yields the best matching score. Results with Llama3.1 and LLaVA-1.6.

| Entail. | Contrad. | Caption | Winoground | | | EqBen | | |
|---|---|---|---|---|---|---|---|---|
| | | | Text | Image | Group | Text | Image | Group |
| ✓ | | | 49.3 | 47.3 | 36.0 | 45.0 | 57.9 | 33.6 |
| | ✓ | | 31.7 | 38.0 | 24.2 | 20.7 | 27.1 | 13.6 |
| ✓ | ✓ | | 46.8 | 57.5 | 39.0 | 47.1 | 60.0 | 38.6 |
| ✓ | ✓ | ✓ | 52.0 | 61.3 | 42.8 | 58.6 | 64.3 | 47.1 |

Table 7: Mixture of VLMs. We show results of balancing the scores of entailments and contradictions ($\alpha_1$) and the given caption ($\alpha_2$) with different models.

| InstructBLIP | LLaVA-1.5 | LLaVA-1.6 | Winoground | | | EqBen | | |
|---|---|---|---|---|---|---|---|---|
| | | | Text | Image | Group | Text | Image | Group |
| ✓ | ✓ | | 49.3 | 55.5 | 41.0 | 48.6 | 58.6 | 40.7 |
| ✓ | | ✓ | 51.5 | 57.3 | 42.5 | 51.4 | 57.9 | 40.0 |
| | ✓ | ✓ | 55.0 | 61.3 | 47.5 | 57.9 | 65.0 | 47.9 |

sively boost the final matching performance. We further explored different combinations to mitigate the "semantic drift" problem by balancing the matching scores from different models. We show in Table 7 how combining different models boosts the final compositional evaluation.

## 7 CONCLUSION

In this work, we introduce Caption Expansion with Contradictions and Entailments (CECE), a principled approach designed to enhance compositional reasoning in vision-language models. CECE leverages Natural Language Inference to generate diverse entailments and contradictions, aimed to expand the semantic understanding of textual descriptions. We conduct extensive evaluations across multiple compositional benchmarks, including Winoground and EqBen, and demonstrate that CECE significantly outperforms prior methods without requiring additional fine-tuning, achieving notable results in alignment with human judgments for text-to-image evaluation. Through comprehensive analysis, we show that CECE enhances interpretability and provides a balanced semantic representation, which is crucial for nuanced image-text matching and reasoning. Our results indicate that combining both entailments and contradictions allows vision-language models to consider both inclusive and exclusive semantic cues, leading to interpretable and less biased compositional reasoning. We encourage future work on interpretable multimodal frameworks that can leverage structured semantic expansions across diverse domains and tasks.

**Broader Impact.** CECE effectively improves the interpretability and robustness of vision-language models, and contributes to fairer AI systems that align more closely with human reasoning, reducing its reliance on superficial correlations. CECE also has the potential to improve a wide range of applications, such as assistive technologies for people with visual impairments, educational tools, and creative content generation. However, like other works that leverage LLMs, CECE also poses risks related to the generation of misleading content or misuse in malicious contexts. The enhanced contextual interpretation and semantic descriptions could be used to make fabricated or altered visual content more convincing, amplifying the risks associated with misinformation. We encourage the responsible use of frameworks like CECE, with an emphasis on transparency, ethical guidelines, and mechanisms for monitoring and mitigating potential misuse.

**Acknowledgments.** We thank the anonymous reviewers as well as the members of the University of Maryland CLIP lab for their thoughtful and thorough feedback. This work was supported by the NSF CAREER Award No. 2339746 (Rudinger). Any opinions, findings, and conclusions or recommendations expressed in this material are those of the author(s) and do not necessarily reflect the views of the National Science Foundation.

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

# A APPENDIX

## A.1 PROMPT DETAILS

In the caption expansion step, we instruct the LLM to generate both entailments and contradictions in a single output rather than in separate steps, as this helps the LLM maintain a balanced semantic context, leading to more coherent and complementary outputs. Since the LLM often tends to output simple negative statements for contradictions (e.g., using words like *"no"* or *"not"*), we explicitly instruct the model to avoid such terms. This ensures that the generated contradictions are more diverse and meaningful, rather than being straightforward negations. In addition, following prior work (Mitra et al., 2024), we instruct the LLM to output the generated entailments and contradictions in *JSON*, as it follows an easy format to parse the outputs. Prompt template is in Figure 4.

```
Prompt Template

Given the sentence:  {Caption}
Think step by step.  What could be entailed?

First, provide two concise descriptions that entail the sentence.
Include attributes mentioned in the sentence (color, size, position,
amounts).  If there is a verb, rephrase it to entail the sentence.
Some possible verb entailments include:
looking -> (participant) person's eyes over something (do not include
the recipient)
kissing -> (participant) person's lips touching someone (do not include
the recipient)
talking -> (participant) person's mouth open (do not include the
recipient)
hugging -> (participant) person's arms reaching the other person
(recipient)
person hitting -> (participant) person is in motion
object hitting -> (participant) object is damaged

The descriptions need to be specific and implicitly entailed in the
sentence.  Include world knowledge and common sense assumptions.
For example:
Sentence:  A yellow unicorn talks to a tall person
Nouns:  plant, unicorn, ball, person, sky
Entailed descriptions:  [Yellow unicorn's mouth is open, Yellow unicorn
is gesturing]

Then, provide the opposite sentence to each sentence, do not include
negations.
For example:
Sentence:  Person talks to unicorn
Nouns:  unicorn, ball, person, sky
Opposite descriptions:  [Tall person's mouth is open, Tall person is
gesturing]

Finally, output the entailed descriptions in a json format.
{
"Entailed descriptions":  [Yellow unicorn's mouth is open, Yellow
unicorn is gesturing]
"Opposite descriptions":  [Tall person's mouth is open, Tall person is
gesturing]
}
```

Figure 4: Prompt template used in Caption Expansion. **{Caption}** is replaced for each sample.

## A.2 Error Analysis

We further examine cases where CECE fails comparing with using original caption only (LLaVA-1.6) and Sentence Decomposition via Semantics (SDS). We show detailed entailments and contradictions in CECE column, where **CECE Final Score** is from Equation 6 with $\alpha_1 = 0.5$ and **Overall Final Score** is from Equation 6 where $\alpha_2 = 0.6$. Note that both $\alpha$ values are kept consistent throughout all our experiments. Figure 5 shows examples where LLaVA-1.6 correctly predicts the score relationship between images given the text but both SDS and CECE fail. Figure 6 shows examples where LLaVA-1.6 and SDS correctly predict the score relationship between images given the text but CECE fails. Figure 7 shows examples where LLaVA-1.6, SDS and CECE all fail. Finally, Figures 8 − 11 show examples where CECE semantically drift from the original caption.

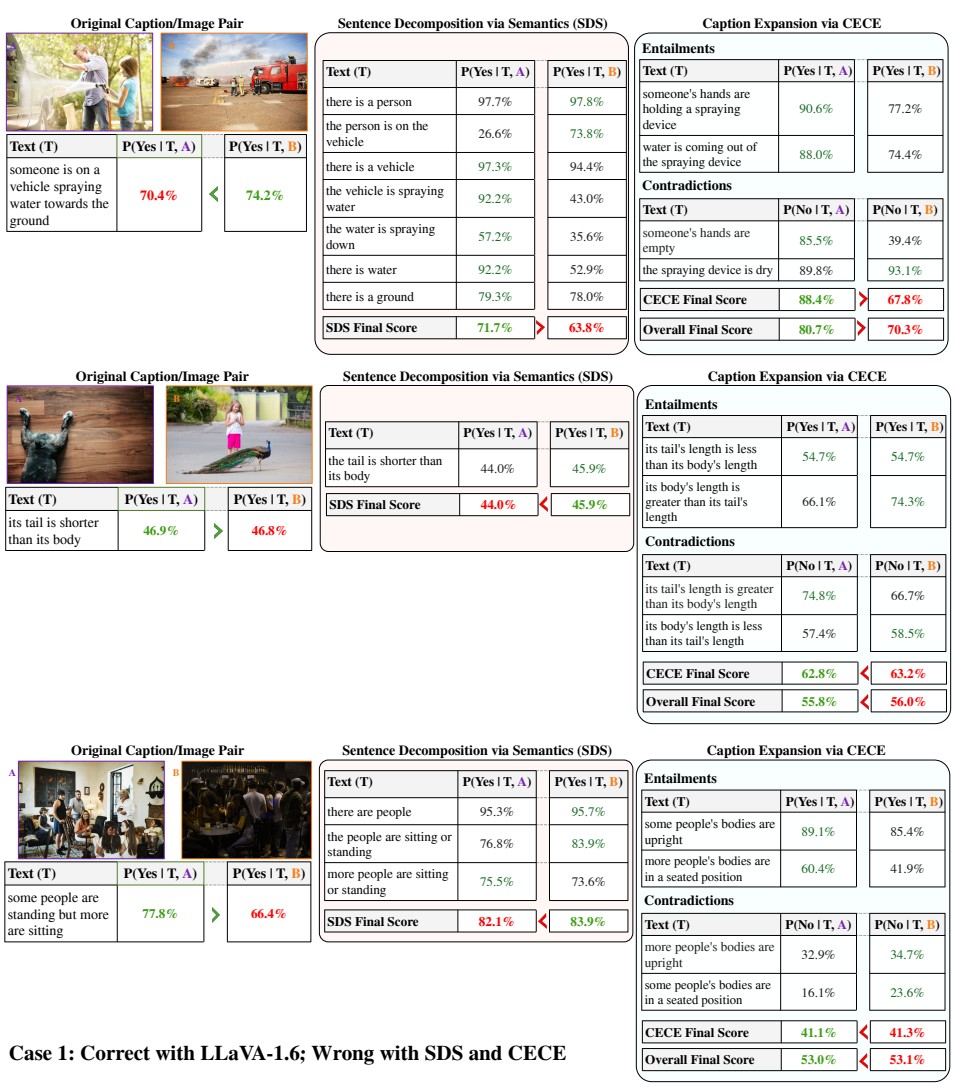

Figure 5: Qualitative error analysis: cases where both SDS and CECE fail: a) In the first case, SDS correctly decomposes the caption and focuses on the *vehicle*, which is the one *spraying water*. On the other hand, through CECE, the LLM incorrectly focuses on a person *spraying water*. However, both cases fail since the VLM is unable to identify the vehicle *spraying water*. b) In the second case, SDS fails to decompose the caption into smaller parts, and repeats the given text. Although CECE produces correct entailments and contradictions, the VLM fails to match the correct image-text pair. c) Similarly, for the third case, the VLM seems to fail to match the correct image, which seems too difficult to parse due to out-of-focus and illumination issues.

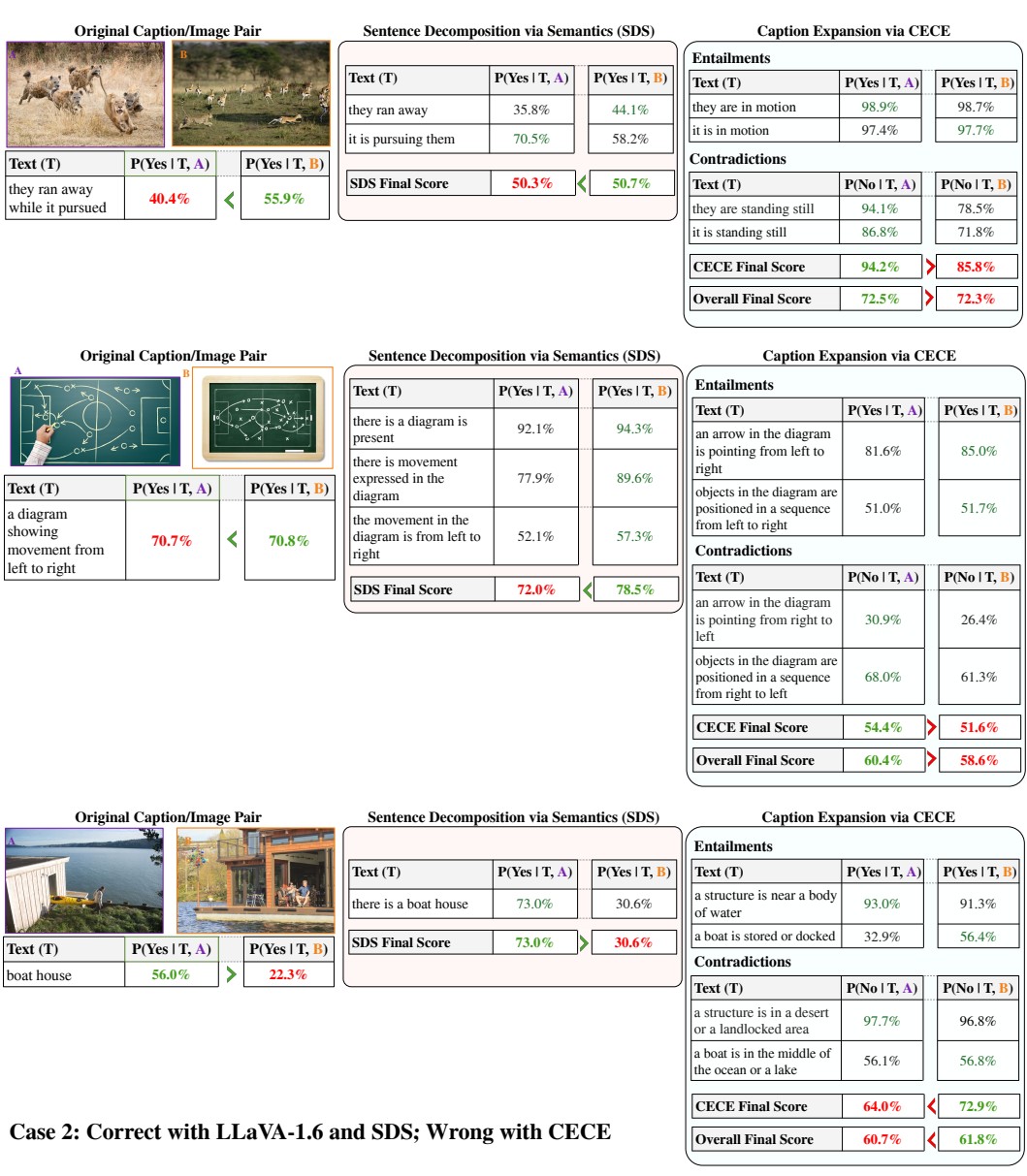

Figure 6: Qualitative error analysis: cases where only CECE fails: a) In the first case, SDS correctly decomposes the caption, although *they* and *it* are not concretely stated (e.g., could refer to animals or objects), breaking the sentence into two separate statements is sufficient for the VLM to match the correct image. On the other hand, the contradictions generated via CECE introduce statements that are correct, but lead to incorrect conclusions given the nature of the data (i.e., still images). b) Similarly, the contradictions generated for the second case introduce incorrect negative statements that weigh over the entailments and lead to the incorrect matching. c) In the third case, SDS is unable to decompose the sentence, and the given description contains the name of the referring object (i.e., *boat house*). Although both entailments and contradictions generated via CECE provide a more detailed or fine-grained set of descriptions, the VLM is unable to identify the correct pair.

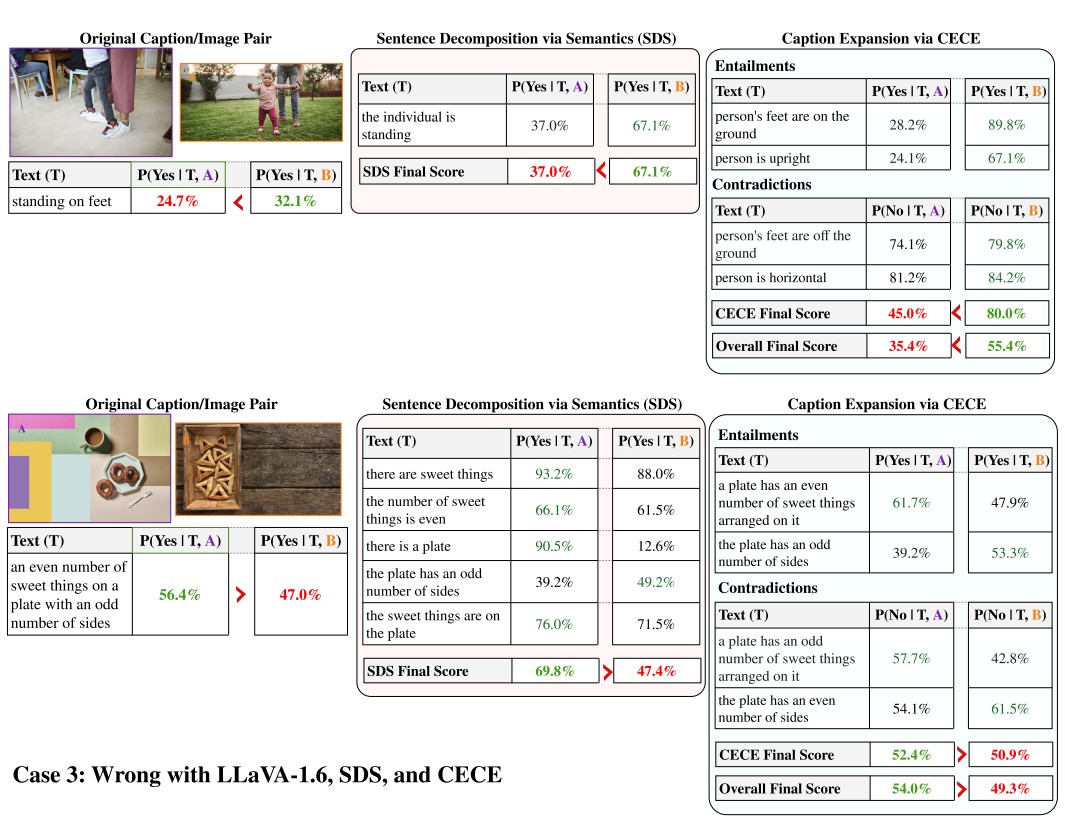

Figure 7: Qualitative error analysis: cases where LLaVA-1.6, SDS and CECE all fail: a) In the first case, while CECE correctly generates entailments and contradictions, the VLM is unable to match the correct image-text pair. In this case, both images are cutout and it may be difficult for the model to identify the people in the scene. Note that SDS is unable to break the sentence. b) As opposed to the previous one, in the second case, SDS is able to correctly decompose the given caption, but the VLM is unable to score the matching image-text pair. CECE on the other hand, fails to generate meaningful entailments and contradictions, leading to an incorrect output. However, the failure margin from CECE is lower than that produced by SDS.

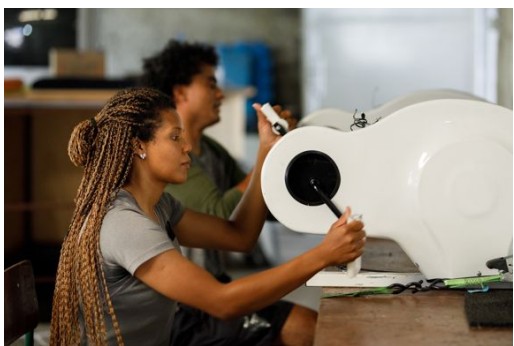 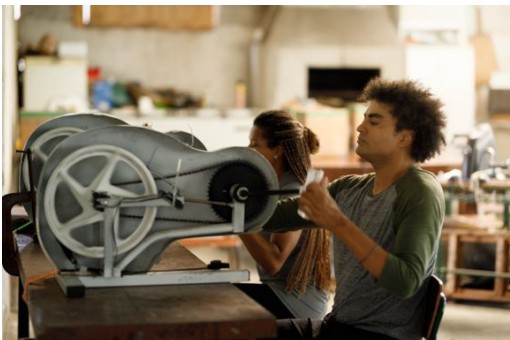

**Caption:** a person with long braids is exercising in front of a person without braids

**Entailments:**
$E_1$: a person with long braids is moving their body.
$E_2$: a person without braids is standing in front of the exercising person.

**Contradictions:**
$C_1$: a person without braids is moving their body.
$C_2$: a person with long braids is standing in front of the exercising person.

**Caption:** a person without braids is exercising in front of a person with long braids

**Entailments:**
$E_1$: a person's body is moving.
$E_2$: a person is in front of another person.

**Contradictions:**
$C_1$: a person with braids is exercising in front of a person without braids.

Figure 8: Qualitative error analysis: cases where CECE semantically drift from the original caption. In this case, the LLM incorrectly outputs *"a person without braids is standing in front of the exercising person"* as an entailment. Similarly, the first two entailments for the second caption do not contain any particular attribute to refer to the right person.

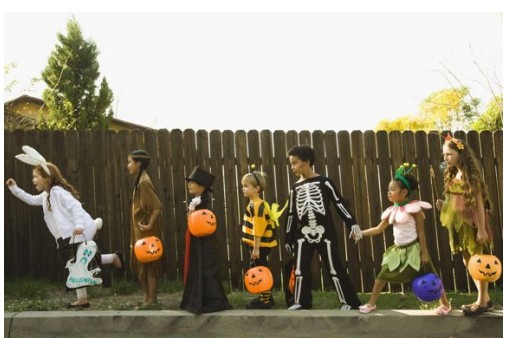 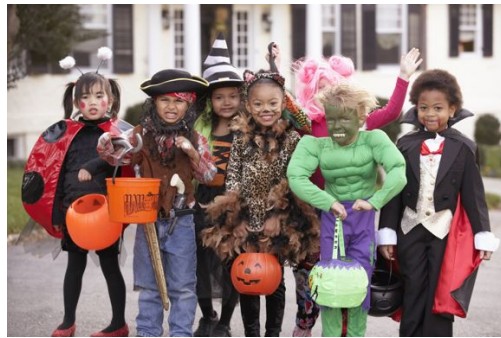

**Caption:** on halloween, a child dressed as an insect is to the right of a child dressed as a mammal

**Entailments:**
$E_1$: a child dressed as an insect is positioned to the right of a child dressed as a mammal.
$E_2$: two children are standing next to each other.

**Contradictions:**
$C_1$: a child dressed as a mammal is to the right of a child dressed as an insect.
$C_2$: a child dressed as a mammal and a child dressed as an insect are standing next to each other with the mammal on the right.

**Caption:** on halloween, a child dressed as a mammal is to the right of a child dressed as an insect

**Entailments:**
$E_1$: a child dressed as a mammal is positioned to the right of a child dressed as an insect.
$E_2$: there are at least two children, one dressed as a mammal and the other as an insect.

**Contradictions:**
$C_1$: a child dressed as an insect is to the right of a child dressed as a mammal.
$C_2$: a child dressed as an insect is positioned to the left of a child dressed as a mammal.

Figure 9: Qualitative error analysis: cases where CECE semantically drift from the original caption. In this case, the LLM provides the incorrect assumption that only two children are standing next to each other, and is also unable to break the caption into smaller semantic components.

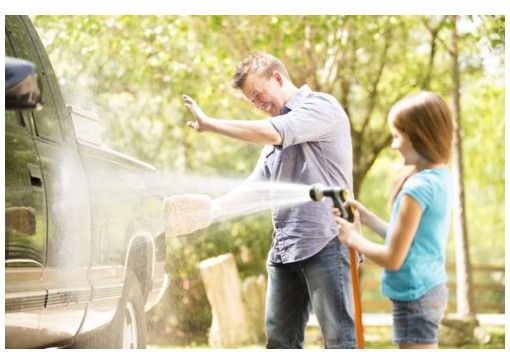 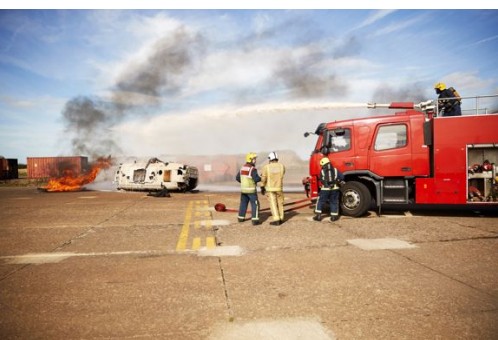

**Caption:** someone on the ground is spraying water towards a vehicle

**Entailments:**

$E_1$: someone's hand is moving with a hose or spray nozzle.
$E_2$: water is coming out of a hose or spray nozzle towards a vehicle.

**Contradictions:**

$C_1$: someone's hand is moving away from a hose or spray nozzle.
$C_2$: water is coming out of a hose or spray nozzle away from a vehicle.

**Caption:** someone is on a vehicle spraying water towards the ground

**Entailments:**

$E_1$: someone's hands are holding a spraying device.
$E_2$: water is coming out of the spraying device.

**Contradictions:**

$C_1$: someone's hands are empty.
$C_2$: the spraying device is dry.

Figure 10: Qualitative error analysis: cases where CECE semantically drift from the original caption. Winoground sample with tag $Both_L$. These are cases where both objects and relations are swapped or a single swap that changes parts of speech. In this case, while the decomposed captions entail and contradict the original caption, they fail to describe the precise action depicted in the image. This happens when the original captions swap multiple objects, attributes, and actions.

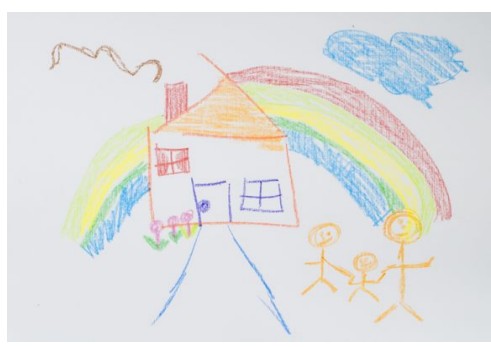 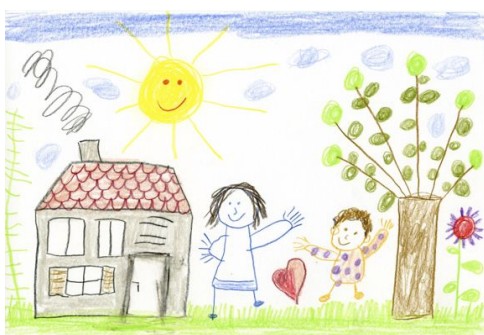

**Caption:** there are three people and two windows

**Entailments:**
$E_1$: three people are present.
$E_2$: two windows are present.
$E_3$: three people are standing or sitting in a room or area
$E_4$: two windows are part of a building or structure

**Contradictions:**
$C_1$: no people are present.
$C_2$: no windows are present.

**Caption:** there are two people and three windows

**Entailments:**
$E_1$: two people are present.
$E_2$: three windows are present.
$E_3$: two people are standing or sitting in a room or a space
$E_4$: three windows are part of a building or a structure

**Contradictions:**
$C_1$: there is one person.
$C_2$: there is one window.

Figure 11: Qualitative error analysis: cases where CECE semantically drift from the original caption. Winoground sample with tag $Symbolic_V$. These samples contain captions for non-natural images, such as drawings or illustrations. In this case, the entailments and contradictions are mostly correct, but the LLM introduces incorrect assumptions about the location of the people present in the image.

