# OpenReview forum: "Natural Language Inference Improves Compositionality in Vision-Language Models"
_ICLR.cc/2025/Conference — ICLR 2025 Poster_

### Official Review · Reviewer_G1JW · 2024-11-03

**Soundness:** 3
**Presentation:** 3
**Contribution:** 3
**Rating:** 6
**Confidence:** 3

**Summary:**

This work proposes to utilize entailments and contradictions generated by LLMs conditioned on captions on visio-linguistic compositionality benchmarks. Specifically, the authors prompt LLMs to generate entailment and contradiction statements in a one-shot manner, then calculate a weighted average of the conditional likelihood of a 'yes/no' response from a VL model, conditioned on image-statement pairs. The statements include the generated entailment/contradiction statement and the original caption. And the weights are hyperparameters. The proposed method shows effectiveness on both EqBen and Winoground benchmarks.

**Strengths:**

- The paper is clear and easy-to-follow.
- The proposed method shows significant gains compared with other baselines on two visio-linguistic compositionality benchmarks.

**Weaknesses:**

- Although the proposed methods demonstrate empirical effectiveness, the motivation for including contradiction statements lacks clarity. According to the prompt in Appendix A.1, the LLM is instructed to generate contradiction statements as sentences opposite to the entailment statements. This setup implies that each pair of entailment and contradiction statements may be semantically equivalent. Given this, it is unclear where the benefit lies in incorporating both entailments and contradictions.
- In Table 6, it would be helpful to include the performance results for using contradictions alone.
- In the qualitative error analysis in Fig. 5, the entailments and contradictions generated by the proposed method generally represent only a subset of the caption’s full semantics, rather than comprehensively capturing its meaning. This limitation may introduce semantic bias. An ablation study on the impact of the number of entailment/contradiction statements, along with an analysis of the semantic diversity among these statements, would enhance the understanding of their effects.

**Questions:**

Refer to weaknesses.

---

> ### Author Response · Authors · 2024-11-22
>
> We thank Reviewer **G1JW** for all the feedback. We are glad you found our paper clear and easy to follow. We address the weaknesses and questions below.
>
>
> **[W1] Why contradictions help.** Natural Language Inference (NLI) tasks inherently require models to leverage common-sense knowledge to determine semantic sub-sets and relations.
> Our experiments show that CECE provides a strong prior for caption expansion and exclusion by considering both entailments and contradictions. Specifically, we instruct the LLM to exclude negations; without this instruction, the LLM would output: *“the banana is _not_ in two or more pieces”* instead of *“the banana is in one piece”* (example from Figure 2), or *“no water is coming out of the spraying device”* instead of *“the spraying device is dry”* (example from Figure 5).
>
> NLI-based expansion produces semantically equivalent statements that contain different vocabulary and sentence structures; thus, the resulting subset of captions (both entailments and contradictions) aids the VLM in covering cases where the caption is too complex or ambiguous.
>
> Contradictions contribute uniquely to this process by offering an additional mechanism for validation. If a given image-caption pair is correct, its contradiction represents an untrue statement of the image. For example, in Figure 2, the correct caption is *“there is a split banana”*. The generated contradiction, *“the banana is in one piece”*, is a description inconsistent with the image. When the VLM correctly identifies the contradiction as untrue, this adds further evidence supporting the correctness of the original image-caption pair. Thus, contradictions enhance compositional reasoning by providing a complementary layer of validation that enriches the semantic understanding of the image-text pairs. Contradictions help the model disambiguate nuanced relationships, such as in the example of *“split banana”* vs. *“banana split.”* This interplay between entailments and contradictions ensures robust coverage of the semantic space, improving the model's performance on complex tasks.
>
>
> **[W2] Include contradictions alone in Table 6.** We include the results when using only the contradictions below and in the revised version. As expected, contradictions alone do not yield state-of-the-art results; however, the result is competitive against prior SDS approaches.
>
>
> | Entail. | Contrad. | Caption | WinoG Text | WinoG Image | WinoG Group | EqBen Text | EqBen Image | EqBen Group |
> |---------|----------|---------|-----------------|------------------|------------------|------------|-------------|-------------|
> | ✓       |          |         | 49.3 | 47.3 | 36.0 | 45.0       | 57.9 | 33.6 |
> |          | ✓        |         | 31.7 | 38.0 | 24.2 | 20.7       | 27.1 | 13.6 |
> | ✓       | ✓        |         | 46.8 | 57.5 | 39.0 | 47.1       | 60.0        | 38.6        |
> | ✓       | ✓        | ✓       | 52.0 | 61.3 | 42.8 | 58.6       | 64.3        | 47.1        |
>
>
> **[W3] Impact of the number of generated captions and semantic diversity.** We take the number of entailments and contradictions for Winoground, and compute some statistics:
>  | **Statistic** | **Entailments** | **Contradictions** |
> |-------------------------|---------------|----------------|
> | Mean | 2.225 | 2.1375 |
> | Standard Deviation | 0.7726 | 0.6234 |
> | Maximum Value | 10 | 9 |
> | Minimum Value | 2 | 1 |
> | Median | 2.0 | 2.0 |
>
>
> Broadly, the LLM generally follows the instruction of 2 entailments and 2 contradictions; however, sometimes it outputs up to 10. In our experiments, we take all outputs from the LLM to compute the final values. We found that, in some cases where the caption is correctly structured and contains more than one action, the LLM tends to provide shorter and more nuanced entailments and contradictions, resulting in more captions.
>
>
> For semantic diversity, we show the Jaccard Similarity to assess the lexical diversity of the generated captions (L428-L431). CECE captions and the original captions show a low similarity score; this supports the diversity statement. We also included human evaluations on captions expansions (L444-L457). Using a Likert scale, the entailment captions received an average score of 4.7, while the contradiction captions received an average score of 1.7. This indicates that the entailment and contradictions are both accurately generated.
>
> Each set of entailments and contradictions provides evidence about the relationship between the image and caption, with some being more discriminative than others in specific cases. This does not necessarily imply a bias but reflects the complementary nature of these expansions in covering diverse semantic dimensions. Entailments often affirm subtle consistencies, while contradictions highlight mismatches that are crucial for distinguishing between closely related visual-text pairs.
>
> We hope our response answers your questions. Please let us know if you would like to ask for any further clarifications.

---

> > ### Author Response · Authors · 2024-12-02
> >
> > Dear Reviewer G1JW,
> >
> > We sincerely thank you for your valuable comments and feedback on our submission. As the rebuttal period is coming to an end, we wanted to follow up to see if you have any additional questions or concerns about our responses. We have also submitted a revised version of our work addressing your feedback. If there’s anything further you’d like us to clarify, we’d be happy to address it. Thank you again for your time and for reviewing our paper.

---

### Official Review · Reviewer_ZcB9 · 2024-11-03

**Soundness:** 3
**Presentation:** 3
**Contribution:** 3
**Rating:** 8
**Confidence:** 4

**Summary:**

The authors propose a new text-image alignment metric, CECE, that considers the original caption alongside a set of LLM-generated entailed and contradictory statements as opposed to past work breaking down captions into smaller sub-statements. This is motivated by a desire to increase lexical diversity and improve performance on captions involving reasoning or world knowledge. The authors validate their approach on Winoground and EqBen and also measure alignment with human judgements on five benchmarks for text-to-image generation and one benchmark for text-to-3D generation, outperforming prior work on all but StanfordT23D. This is coupled by finer-grained analyses of performance, lexical diversity and correctness on Winoground.

**Strengths:**

1) The method achieves state-of-the art performance on Winoground and EqBen, outperforming the second best evaluated model by large margins of 11.2% and 9.3% respectively. Its alignment with human judgements is similarly strong, albeit with a lesser gap over prior work.
2) This improvement is achieved via a relatively simplistic inference-time strategy requiring no finetuning.
3) It was unexpected for me that "expanding" a caption via entailed and contradictory statements would lead to better performance than breaking down statements into smaller components as in prior work. This could inspire further inference-time strategies either refining the current approach or combining it with elements of past work.

**Weaknesses:**

1) One unaddressed limitation of the approach compared to the VQAScore baseline is the inference-time cost of additionally using Llama3.1 70B for caption expansion. This is an integral component of the approach and I would have expected ablations of the LLM (for instance by using smaller variants of Llama3.1) in the same way that there were ablations of the VLM.

2) I think the paper would be improved by further discussions of when CECE underperforms against the baselines. On Table 4, for instance, CECE without ensembling underperforms against the VQAScore baseline for the Text score for the Object_L and Both_L categories (and by a roughly 25% margin for the latter). Similarly, for the Pragmatic_V split, CECE can only decisively outperform the VQAScore baseline after ensembling and is either competitive with or poorer than the VQAScore baseline otherwise. An analysis of why would lead to a better understanding of the strengths and weaknesses of the approach and would help future work improve the method further.

3) On a related note, I think the comparisons for the ensembling strategy would be fairer if a similar ensembling strategy were applied for VQAScore and DSG, particularly when considering the boost ensembling affords on Table 4. For instance, would VQAScore still perform worse if some weighted average of the Llava-1.5 and 1.6 probabilities were used instead?

**Questions:**

1) In most of the Winoground splits on Table 4, CECE performs better on selecting images rather than text unlike the VQAScore baselines. Any ideas why this may be the case?

2) Is there any reason why CECE with InstructBLIP was not included for Tables 2 and 3 (especially considering that VQAScore with InstructBLIP achieves superior performance compared to Llava-1.5)?

3) Is the value of M for the number of entailments and contradictions set in stone or variable based on the input caption?

---

> ### Author Response · Authors · 2024-11-22
>
> We thank Reviewer **ZcB9** for all the feedback. We are glad you found our method simple yet effective, as it does not require fine-tuning since it is performed in inference time. We also hope our work could inspire further inference-time strategies, building on our findings.
> We address the weaknesses and questions below.
>
> **[W1] Ablations of the LLM.** We include additional experiments using Llama 3.1-8B. This was also pointed out by reviewer *Xyex*, and we thank both for bringing it to our attention. We include these experiments in the revised version of the manuscript.
> We also show a fraction of the table below for reference.
>
> | Method                           | Tools                      | LLM              | Winground Text | Winground Image | Winground Group  | EqBen Text | EqBen Image | EqBen Group |
> |----------------------------------|----------------------------|------------------|----------------|-----------------|------------------|------------|-------------|-------------|
> | Random Chance                    | -                          | -                | 25.0           | 25.0            | 16.7             | 25.0       | 25.0        | 16.7        |
> | Human Evaluation                 | -                          | -                | 89.5           | 88.5            | 85.5             | -          | -           | -           |
> | *End-to-end models*            |                            |                  |                |                 |                  |            |             |             |
> | VQAScore (Lin et al., 2024)      | LLaVA-1.5                  | -                | 45.5           | 41.3            | 29.8             | 45.0       | 47.1        | 28.6        |
> | VQAScore (Lin et al., 2024)      | LLaVA-1.6                  | -                | 46.8           | 45.8            | 31.3             | 46.4       | 54.3        | 32.9        |
> | *Sentence Decomposition via Semantics (SDS)* |              |                  |                |                 |                  |            |             |             |
> | DSG (Choe et al., 2023)          | LLaVA-1.5                  | Llama3.1 (8B)    | 5.7            | 9.5             | 3.7              | 10.0       | 14.3        | 6.4         |
> | DSG (Choe et al., 2023)          | LLaVA-1.6                  | Llama3.1 (8B)    | 4.5            | 10.2            | 2.7              | 10.7       | 14.3        | 6.4         |
> | *Ours*            		   |                            |                  |                |                 |                  |            |             |             |
> | CECE                             | LLaVA-1.5                  | Llama3.1 (8B)    | 47.7           | 49.7            | 35.5             | 48.6       | 54.3        | 35.0        |
> | CECE                             | LLaVA-1.6                  | Llama3.1 (8B)    | 48.0           | 57.5            | 38.7             | 50.7       | 64.3        | 40.0        |
> | CECE                             | LLaVA-1.5, LLaVA-1.6       | Llama3.1 (8B)    | 50.0           | 53.5            | 39.0             | 53.6       | 57.1        | 40.7        |
>
>
> Our results show that even with a significantly smaller model, our CECE approach consistently outperforms the end-to-end approaches, including the GPT4-Vision model (which leverages the GPT4V-Eval method proposed in Zhang et al., 2023).
>
> As we mentioned in the Reviewer’s *Xyex* response, it is also worth noting that Llama3.1 8B outputs hurt other Sentence Decomposition via Semantics (SDS) methods. Our experiments show that a smaller LLM tends to output unrelated or truncated outputs with these approaches. We hypothesize that this behavior is due to the complexity of the prompt and required task decomposition in these methods (e.g., dividing the sentence as a subset of tasks that require separate processing, along with the larger textual context due to the multi-shot example demonstrations).

---

> ### Author Response · Authors · 2024-11-22
>
> **[W2] Why the Text Score is better in Both_L, Symbolic_V baselines.**
> Thank you for raising this question! Upon further investigation, we identified an error in the reported results. The baseline values for LLaVA-1.5 were slightly overestimated, while the CECE values with LLaVA-1.6 included only the entailment scores, leading to a slight underestimation. We updated Table 4 in the manuscript with the correct scores and attach the table below for reference:
>
>
> | Method               | Object_L Text    | Object_L Image     | Object_L Group     | Relation_L Text     | Relation_L Image     | Relation_L Group     | Both_L Text       | Both_L Image       | Both_L Group       | Symbolic_V Text     | Symbolic_V Image     | Symbolic_V Group     | Pragmatics_V Text     | Pragmatics_V Image     | Pragmatics_V Group     |
> |----------------------|--------------------|---------------------|---------------------|----------------------|-----------------------|-----------------------|--------------------|---------------------|---------------------|----------------------|-----------------------|-----------------------|------------------------|------------------------|------------------------|
> | Human               | 92.20             | 90.78              | 88.65              | 89.27               | 90.56                | 86.70                | 76.92             | 57.69              | 57.69              | 96.43               | 92.86                | 92.86                | 58.82                 | 41.18                 | 41.18                 |
> | InstructBLIP        | 42.5              | 49.7               | 27.7               | 34.3                | 33.9                 | 20.2                 | **65.4**          | 38.5               | 34.6               | 31.7                | 21.9                 | 14.6                 | 25.0                  | 29.2                  | 8.3                   |
> | LLaVA-1.5           | 46.1              | 46.8               | 28.4               | 45.1                | 48.3                 | 30.8                 | **65.4**          | 46.2               | 38.5               | 46.3                | 31.7                 | 20.8                 | 32.9                  | 41.7                  | 12.5                  |
> | LLaVA-1.6           | 48.2              | 53.9               | 35.5               | 43.8                | 40.8                 | 27.9                 | **65.4**          | 46.1               | 38.5               | 46.3                | 41.5                 | 26.8                 | 39.2                  | 41.7                  | 16.7                  |
> | LLaVA-1.5+1.6       | 51.7              | 53.9               | 36.2               | 46.2                | 44.2                 | 34.3                 | 61.5              | 46.1               | 38.5               | 56.1                | 43.9                 | 31.7                 | 37.5                  | 33.3                  | 20.8                  |
> |  | | | | | | | | | | | | | | | |
> | DSG                 | 45.4              | 44.0               | 27.6               | 42.9                | 41.2                 | 30.9                 | 61.5              | 53.8               | 46.1               | 46.3                | 41.5                 | 29.3                 | 33.3                  | 33.3                  | 20.8                  |
> | DSG*                | 52.5              | 50.3               | 35.5               | 51.0                | 45.5                 | 34.3                 | 53.8              | 42.3               | 34.6               | 41.4                | 31.7                 | 19.5                 | **50.0**              | 41.7                  | **33.3**              |
> | *Ours* | | | | | | | | | | | | | | | |
> | CECE         | 51.8              | 66.0               | 43.3               | 51.5                | **59.2**             | 42.5                 | 57.7              | 38.3               | 42.3               | 53.7                | **65.9**             | 39.0                 | 45.8                  | **50.0**              | **33.3**              |
> | CECE*        | **56.7**          | **68.8**           | **49.7**           | **53.6**            | 56.7                 | **45.9**             | 57.7              | **61.5**           | **50.0**           | **58.5**            | 58.5                 | **43.9**             | 37.5                  | 41.7                  | **33.3**              |
>
>
>
> The text score measures if the model can select the correct caption given an image.
> In our manual evaluation, we observed that the LLM outputs tend to be noisy when the original captions contain multiple objects, attributes, and actions. These captions are generally tagged as *Both_L* (the subset of samples where both objects and relations are swapped, or a single swap that changes parts of speech).

---

> ### Author Response · Authors · 2024-11-22
>
> **[W3] Ensembling VQAScore Llava1.5+Llava1.6: Table 4.** We show results when ensembling VQAScores with LLaVA-1.5 and LLaVA-1.6. In this case, only the *Symbolic_V* tag subset surpasses our CECE without LLaVA-1.5 and LLaVA-1.6 ensembling.
> *Symbolic_V* tag includes samples with non-natural images such as drawings or illustrations. While the LLM outputs are aligned with common-sense knowledge, drawings are not necessarily grounded in reality. For the samples tagged as *Pragmatics_V*, the images need to be interpreted non-literally due to idiomatic uses of language in a caption.
>
> In addition, we add qualitative samples in the appendix (Figures 10 and 11). These observations align with the semantic drift issue we mitigate by balancing CECE with the whole caption score. We include additional examples in the appendix (please also take a look at Figures 8 and 9, which are closely related).
>
> **[Q1] Why CECE performs better Image Score.** The text score evaluates the model's ability to distinguish between captions for a given image, and the image score evaluates its ability to distinguish between images for a given caption. Prior work has pointed out that some visual features may be subtle or difficult for the models to capture at their input resolutions, making the image score task harder [2]. This is particularly true for almost all end-to-end models (e.g., in [1], all models shown in Table 3 perform worse under the image score metric).
>
> A key advantage of the CECE and SDS approaches is their interpretability. By expanding or decomposing captions into smaller semantic units, these methods provide models with more fine-grained information to match against images. With CECE, noise or errors in the expansion process for one caption apply equally to both images, maintaining consistency in the scoring process for the image score case. In contrast, the text score involves comparing two captions with separate sets of entailments and contradictions against one image, which introduces variance and increases the likelihood of errors or inconsistencies affecting the result. This asymmetry makes the image score less sensitive to erroneous expansions, justifying CECE’s stronger performance in this metric.
>
> **[Q2] Including CECE with InstructBLIP in Tables 2 and 3:** We run experiments with CECE using InstrucBLIP as the VLM backbone. Our results show that CECE outperforms the base model, and particularly in Table 3, CECE outperforms both InstructBLIP and LLaVA-1.5. We include both updated tables in the revised version. We also add Table 3 below for reference.
>
>  | Method                                           | Pairwise Acc | Pearson | Kendall |
> |--------------------------------------------------|--------------|---------|---------|
> | **End-to-end models**                            |              |         |         |
> | CLIPScore                                        | 61.0         | 48.1    | 32.6    |
> | BLIPv2Score                                      | 56.6         | 34.3    | 23.4    |
> | InstructBLIP                                     | 68.0         | 59.5    | 47.5    |
> | LLaVA-1.5                                        | 64.9         | 55.8    | 40.8    |
> | *Finetuned on human feedback*                  |              |         |         |
> | ImageReward                                      | 66.3         | 57.1    | 43.6    |
> | PickScore | 60.1         | 41.3    | 30.3    |
> | HPSv2 | 55.9         | 31.5    | 21.9    |
> | *CECE* |              |         |         |
> | w/ InstructBLIP | **68.5**  | **64.0**    | **48.4**    |
> | w/ LLaVA1.5 | 65.3         | 57.4    | 41.8    |
>
> **[Q3] Number of entailments and contradictions.** We take the number of entailments and contradictions for Winoground, and compute some statistics:
>  | **Statistic** | **Entailments** | **Contradictions** |
> |-------------------------|---------------|----------------|
> | Mean | 2.225 | 2.1375 |
> | Standard Deviation  | 0.7726 | 0.6234 |
> | Maximum Value | 10 | 9  |
> | Minimum Value | 2 | 1  |
> | Median | 2.0 | 2.0  |
>
> Broadly, the LLM follows the instruction of 2 entailments and 2 contradictions; however, sometimes it outputs up to 10. In our experiments, we take all outputs from the LLM to compute the final values. We found that, in some cases where the caption is correctly structured and contains more than one action, the LLM tends to provide shorter and more nuanced entailments and contradictions, resulting in more captions.
>
> We hope our response answers your questions. Please let us know if you would like to ask for any further clarifications.
>
> ________________
>
> [1] Thrush, Tristan, et al. "Winoground: Probing vision and language models for visio-linguistic compositionality." Proceedings of the IEEE/CVF Conference on Computer Vision and Pattern Recognition. 2022.
>
> [2] Diwan, Anuj et al. “Why is Winoground Hard? Investigating Failures in Visuolinguistic Compositionality.” Conference on Empirical Methods in Natural Language Processing (2022).

---

> > ### Comment · Reviewer_ZcB9 · 2024-11-24
> >
> > I thank the authors for their response, which addresses all of my questions and concerns.

---

> > > ### Comment · Reviewer_ZcB9 · 2024-11-27
> > >
> > > I want to write a follow-up comment to my original response. I am leaning towards accepting the paper due to the robustness of the method to smaller LLMs, the correction of the results in Table 4 and the added results for the SugarCrepe dataset and have accordingly increased my score to an 8.

---

> > > > ### Author Response · Authors · 2024-12-02
> > > >
> > > > Thank you very much for your detailed feedback and for taking the time to review the updates. We also appreciate your time to follow-up on your original response! Your suggestions are invaluable in improving our work, and we sincerely appreciate your recognition of our efforts to address your concerns.

---

### Official Review · Reviewer_1DuW · 2024-11-04

**Soundness:** 3
**Presentation:** 3
**Contribution:** 3
**Rating:** 6
**Confidence:** 3

**Summary:**

This paper proposes Caption Expansion with Contradictions and Entailments (CECE) to enhance compositional reasoning capabilities in vision-language models (VLMs) by leveraging a large language model (LLM) to generate entailments and contradictions through natural language inference (NLI) prompts. The core idea is to prompt the LLM to expand each image caption by producing semantically related statements, both for entailments and contradictions. The intuition is that CECE provides the VLM with lexically diverse cues that encourage deeper visual-textual alignment for complex compositional reasoning. The authors conducted extensive evaluations on two common compositionality benchmarks: Winoground and EqBen, CECE achieves competitive results.

**Strengths:**

This paper is well-written and relatively easy to follow. The core ideas are intuitive. I think The proposed method, CECE, addresses a critical limitation in vision-language models, as vanilla pretrained VLMs have traditionally struggled with compositional reasoning tasks. CECE provides a novel solution that enriches the semantic understanding of VLMs and I see it beneficial to the broader community. The experimental results are competitive with clear improvements over existing methods.

**Weaknesses:**

A crucial limitation of this method is its reliance on two balancing hyperparameters. While the authors provide some analysis, the need for the balancing hyperparmeters could hurt CECE’s generalizability to new tasks and datasets. I think documenting example cases of semantic drift in the appendix would be valuable to the community and it will provide insight into how auxiliary prompting can affect semantic meaning of original captions.

**Questions:**

Minor experimental detail:

Did the authors use any system prompt or chat template for the open-source LLMs during the experiments?

---

> ### Author Response · Authors · 2024-11-22
>
> We thank Reviewer **1DuW** for all the feedback. We are glad you found our core idea intuitive, pointing out CECE as a novel solution that enriches semantic understanding and our experimental results competitive, with clear improvements over existing methods.
> We address the weaknesses and questions below.
>
>
> **[W1] Semantic drift with CECE.** We add detailed explanations for cases where CECE captions drift away from the original semantic meaning in the appendix. Particularly, in Figures 8 and 9 (in the Appendix), we add examples where the LLM outputs imprecise entailments, or the caption is not decomposed appropriately. We also add detailed descriptions in Figures 5-6 to explain cases where CECE fails in the scoring process.
>
>
> **[Q1] Minor experimental detail.** System prompt: we do not use a particular instruction for the system (i.e., an empty string). Chat template: we do not modify the chat template (i.e., *tokenizer.apply_chat_template* has the default functionality from the huggingface model card).
>
>
> We hope our response answers your questions. Please let us know if you would like to ask for any further clarifications.

---

> > ### Author Response · Authors · 2024-12-02
> >
> > Dear Reviewer 1DuW,
> >
> > We sincerely thank you for your valuable comments and feedback on our submission. As the rebuttal period is coming to an end, we wanted to follow up to see if you have any additional questions or concerns about our responses. We have also submitted a revised version of our work incorporating all reviewers' feedback. If there’s anything further you’d like us to clarify, we’d be happy to address it. Thank you again for your time and for reviewing our paper.

---

### Official Review · Reviewer_Xyex · 2024-11-05

**Soundness:** 3
**Presentation:** 3
**Contribution:** 3
**Rating:** 8
**Confidence:** 4

**Summary:**

The paper presents Caption Expansion with Contradictions and Entailments (CECE), a new approach to improve compositionality in the vision-language models by transforming the caption into a natural language inference task. The main idea is to paraphrase the caption into contradictions and entailments with a large language model and then combine them with a vision-language model for final prediction. The results on two important benchmarks (image-to-text and text-to-image) show that CECE outperforms existing baselines without any fine-tuning.

**Strengths:**

The paper is well-written, and the motivation and method are clearly explained. The related work is thorough, which makes it easy to understand the existing body of work on the topic.

The method focuses on manipulating the captions and shows improved performance without any fine-tuning, which is impressive and useful.

The analysis shows that all the components — caption, entailment, and contradiction — are important for the highest performance on the Winoground and EqBen datasets.

**Weaknesses:**

**Inference cost.**
The main weakness of the method is that it requires a very powerful LLM, such as LLama 3.1 70B, for caption expansion. This adds to the inference and compute cost for generating lexically diverse captions. It would be helpful to know if a smaller model can improve performance similarly. It would be ideal to have an experiment showing the performance of the VLM on a range of language models.


**More datasets.**
The paper experiments on two datasets, namely, WinoGround and EqBen. However, Winoground is a small dataset with only 1600 examples. It would be helpful to know if the method works on larger datasets such as SugarCrepe [1] or larger synthetic datasets such as the concept binding dataset [2].


[1] SugarCrepe: Fixing Hackable Benchmarks for Vision-Language Compositionality. NeurIPS: Datasets and Benchmarks Track, 2023.

[2] Does clip bind concepts? probing compositionality in large image models. EACL Findings, 2024.

**Questions:**

See the weaknesses.

---

> ### Author Response · Authors · 2024-11-22
>
> We thank Reviewer **Xyex** for all the provided feedback. We are glad you find our paper well written, with a clear motivation and method. We are also glad you acknowledge CECE’s strong performance, especially since it’s architecture-agnostic and does not require any fine-tuning.
> We address the weaknesses and questions below.
>
> **[W1] Inference cost.** We include results using a smaller open-sourced model, i.e., Llama 3.1-8B in the revised version. For a fair comparison, we also run DSG using this same LLM. We show a fraction of the table below for reference.
>
> | Method                           | Tools                      | LLM              | Winground Text | Winground Image | Winground Group  | EqBen Text | EqBen Image | EqBen Group |
> |----------------------------------|----------------------------|------------------|----------------|-----------------|------------------|------------|-------------|-------------|
> | Random Chance                    | -                          | -                | 25.0           | 25.0            | 16.7             | 25.0       | 25.0        | 16.7        |
> | Human Evaluation                 | -                          | -                | 89.5           | 88.5            | 85.5             | -          | -           | -           |
> | *End-to-end models*            |                            |                  |                |                 |                  |            |             |             |
> | VQAScore (Lin et al., 2024)      | LLaVA-1.5                  | -                | 45.5           | 41.3            | 29.8             | 45.0       | 47.1        | 28.6        |
> | VQAScore (Lin et al., 2024)      | LLaVA-1.6                  | -                | 46.8           | 45.8            | 31.3             | 46.4       | 54.3        | 32.9        |
> | *Sentence Decomposition via Semantics (SDS)* |              |                  |                |                 |                  |            |             |             |
> | DSG (Choe et al., 2023)          | LLaVA-1.5                  | Llama3.1 (8B)    | 5.7            | 9.5             | 3.7              | 10.0       | 14.3        | 6.4         |
> | DSG (Choe et al., 2023)          | LLaVA-1.6                  | Llama3.1 (8B)    | 4.5            | 10.2            | 2.7              | 10.7       | 14.3        | 6.4         |
> | *Ours*            		   |                            |                  |                |                 |                  |            |             |             |
> | CECE                             | LLaVA-1.5                  | Llama3.1 (8B)    | 47.7           | 49.7            | 35.5             | 48.6       | 54.3        | 35.0        |
> | CECE                             | LLaVA-1.6                  | Llama3.1 (8B)    | 48.0           | 57.5            | 38.7             | 50.7       | 64.3        | 40.0        |
> | CECE                             | LLaVA-1.5, LLaVA-1.6       | Llama3.1 (8B)    | 50.0           | 53.5            | 39.0             | 53.6       | 57.1        | 40.7        |
>
> Our results show that even with a significantly smaller model, our CECE approach consistently outperforms the end-to-end approaches, including the GPT4-Vision model (which leverages the GPT4V-Eval method proposed in Zhang et al., 2023). It is also worth noting that Llama3.1 8B outputs hurt other Sentence Decomposition via Semantics (SDS) methods.
>
> Our experiments show that a smaller LLM tends to output unrelated or truncated outputs with these approaches. We hypothesize that this behavior is due to the complexity of the prompt and required task decomposition in these methods (e.g., dividing the sentence as a subset of tasks that require separate processing, along with the larger textual context due to the multi-shot example demonstrations).

---

> ### Author Response · Authors · 2024-11-22
>
> **[W2] More datasets.** We show results using CECE variants with *SugarCrepe* [1]. It is worth noting that prior work that reports strong results in this benchmark tends to perform poorly on Winoground and EqBen. CF2C uses GPT-4V, where the input is an image and the pair of captions, and the model outputs which caption is more likely to describe the image. We include this number in the table below, even though we evaluate using only one image and one caption at a time.
>
>
> | Method             | Tools     | LLM  | Replace Object | Replace Attribute | Replace Relation | Swap Object | Swap Attribute | Add Object | Add Attribute | Avg   |
> |--------------------|-------------------------|------------------|----------------|-------------------|------------------|-------------|----------------|------------|---------------|-------|
> | Human Evaluation   | -                       | -                | 100            | 99                | 97               | 99          | 100            | 99         | 99            | 99.1  |
> | VQAScore           | LLaVA-1.5               | -                | 82.0           | 86.4              | 89.6             | 85.7        | 85.6           | 91.4       | 85.5          | 86.7  |
> | VQAScore           | LLaVA-1.6               | -                | 92.9           | 90.1              | 92.1             | 90.6        | 89.0           | 93.7       | 89.4          | 91.0  |
> | CF2C [2]               | GPT-4V                  | -                | 96.3           | 93.5              | 90.3             | 83.1        | 90.1           | 91.6       | 91.8          | 90.6  |
> | | | | | | | | | | | |
> | DSG                | LLaVA-1.5               | Llama-3.1        | 97.7           | 92.1              | 78.9             | 82.0        | 86.3           | 98.2       | 93.1          | 90.7  |
> | DSG                | LLaVA-1.6               | Llama-3.1        | 97.6           | 93.8              | 80.6             | 83.2        | 91.1           | 98.0       | 96.0          | 91.6  |
> | | | | | | | | | | | |
> | CECE               | LLaVA-1.5               | Llama-3.1        | 93.5           | 88.1              | 88.6             | 85.2        | 81.7           | 91.9       | 84.5          | 87.2  |
> | CECE               | LLaVA-1.6               | Llama-3.1        | 96.6           | 93.0              | 93.3             | 88.5        | 88.6           | 95.5       | 89.3          | 91.8  |
> | CECE               | LLaVA-1.6, LLaVA-1.6    | Llama-3.1 	  | 96.1     	   | 93.0              | 93.8             | 90.2        | 90.8           | 95.9       | 90.2          | 92.6  |
>
> These results provide strong evidence that supports the effectiveness of our work. We expect future research directions that leverage caption expansion techniques (i.e., entailments and contradictions) to develop inference-time methods and training strategies that improve compositional reasoning in vision-language models. We hope our response answers your questions. Please let us know if you would like to ask for any further clarifications.
>
> [1] SugarCrepe: Fixing Hackable Benchmarks for Vision-Language Compositionality. NeurIPS: Datasets and Benchmarks Track, 2023.
>
> [2] https://github.com/RAIVNLab/sugar-crepe/tree/main/gpt-4v-results

---

> > ### Comment · Reviewer_Xyex · 2024-11-25
> > **Response to the authors**
> >
> > Thanks for the response and the additional results. The results on SugarCrepe look impressive. It would be awesome if you could include these results in the paper.
> >
> > Increasing my score to 8 and confidence to 4.

---

> > > ### Author Response · Authors · 2024-12-02
> > >
> > > Thank you very much for your detailed feedback and for taking the time to review the updates. Your suggestions are invaluable in improving our work. We sincerely appreciate your recognition of our efforts to address your concerns. We will include the results on SugarCrepe in the re-revised version.

---

### Meta-Review · Area_Chair_Dwen · 2024-12-20

**Metareview:**

CECE improves compositional reasoning in vision-language models (VLMs) by using a large language model (LLM) to generate entailments and contradictions of captions. These expanded captions provide more diverse cues for better visual-textual alignment, leading to improved performance on compositional tasks like Winoground and EqBen, often without fine-tuning.

The paper is well-written, and the motivation and method are clearly explained. The core ideas are intuitive and the experiment results demonstrate the merit of the work.

**Additional Comments On Reviewer Discussion:**

The reviewers agrees the novelty and contribution of the work. The authors also addressed the concerns of the reviewers by adding more experimental results and analysis.

---

### Decision · Program_Chairs · 2025-01-22

Accept (Poster)